# Heat Inactivation of Methicillin-Resistant *Staphylococcus aureus* Strains from German Dairy farms in Colostrum and Raw Milk

**DOI:** 10.3390/ani13223549

**Published:** 2023-11-17

**Authors:** Mirka E. Wörmann, Ashwini Bhatte, Heidi Wichmann-Schauer, Bernd-Alois Tenhagen, Tobias Lienen

**Affiliations:** Department of Biological Safety, German Federal Institute for Risk Assessment, 10589 Berlin, Germany; bhatte.ashwini@gmail.com (A.B.); heidi.wichmann-schauer@bfr.bund.de (H.W.-S.); bernd-alois.tenhagen@bfr.bund.de (B.-A.T.); tobias.lienen@bfr.bund.de (T.L.)

**Keywords:** MRSA, thermal inactivation, milk, calf, one health

## Abstract

**Simple Summary:**

Methicillin-resistant *Staphylococcus aureus* (MRSA) is a mastitis pathogen in dairy cows. It is common on dairy farms to feed calves milk that cannot be marketed for human consumption for various reasons. Thus, one possible route of MRSA transmission into young stock is via the feeding of contaminated colostrum or raw milk. The aim of our study was to evaluate whether heat treatment of colostrum or raw milk prior to feeding will eliminate MRSA from the colostrum/raw milk and therefore reduce the risk of introducing MRSA into the calf population during the feeding process. We demonstrate that heating colostrum or raw milk at 60 °C causes a substantial reduction in MRSA in these two matrices. However, depending on the MRSA concentration, it may not be sufficient to eliminate all viable MRSA and heat-resistant MRSA can multiply again. Thus, heated colostrum and raw milk should be fed to the calves shortly after the treatment to avoid re-growth of viable MRSA.

**Abstract:**

Methicillin-resistant *Staphylococcus aureus* (MRSA) may cause difficult-to-treat infections in dairy cattle. One possible route of MRSA transmission into calves is via the feeding of contaminated waste milk. We tested the heat resistance of 17 MRSA strains isolated from German dairy farms in colostrum and raw milk in a laboratory approach. Heating colostrum or raw milk at 60 °C for 30 min eliminated all viable MRSA in the milk, provided the MRSA inoculation rate is low (10^3^ cfu mL^−1^). In contrast, raw milk highly inoculated with MRSA (10^6^ cfu mL^−1^) required a holding time of at least 30 min at 70 °C to fully eliminate MRSA from it. However, quantitative analysis showed that a heat treatment for 10 min at 60 °C already significantly reduced the number of viable MRSA in highly inoculated raw milk. Heating colostrum and raw milk above 60 °C may destroy immunoglobulins which are crucial for the calf’s health. Therefore, we suggest that colostrum and raw milk that is to be fed to calves on MRSA-positive dairy farms is heated at 60 °C for at least 10 min to reduce the likelihood of transmitting MRSA. In addition, the 60 °C heat-treated colostrum/raw milk should be fed to the calves as soon as possible to avoid re-growth of viable MRSA.

## 1. Introduction

Methicillin-resistant *Staphylococcus aureus* (MRSA) has been repeatedly isolated from dairy farms worldwide [1,2]. Livestock-associated (LA)-MRSA belonging to the sequence type (ST) 398 are the predominant MRSA found in animal farms in Europe and can be transmitted to humans by animal contact [3,4,5]. Typically, LA-MRSA ST398 lack virulence genes linked to human disease; thus, the risk for severe infections in humans caused by LA-MRSA ST398 is considered to be low [6,7]. However, LA-MRSA ST398 are among the leading pathogens causing bovine mastitis [8,9]. Bovine mastitis is a major concern on dairy farms resulting in substantial financial losses due to discarded milk, treatment costs and premature culling [9,10,11]. Treatment of mastitis commonly involves the use of beta-lactam antibiotics, a class of antibiotics to which MRSA are resistant due to the expression of a modified penicillin-binding protein (PBP2a) encoded by the *mecA* or *mecC* genes [8]. Some authors suggest that the application of large-scale beta-lactam antibiotics on dairy farms may even favor the proliferation of resistant *S. aureus* by creating an environment that suppresses competitors [12,13]. Considering the challenges in treating MRSA mastitis and the low cure rate, it is important to identify transmission routes and work out intervention strategies to reduce the spread of MRSA within dairy cattle herds. MRSA have been repeatedly found in raw milk drawn from individual animals and bulk tank milk (BTM) [2,14,15]. Milk in the mammary gland is generally sterile, unless there is an infection in the udder tissue. Contamination can occur during or after milking due to poor hygienic practice, such as improper use of udder towels and gloves when handling animals and insufficient cleaning of milking equipment [1,16]. It is a common practice on dairy farms to feed unsalable (waste) milk, e.g., milk from treated cows and post-parturient milk, to calves [17]. Thus, a possible route for MRSA transmission from cow to calf is via the feeding of contaminated milk [18,19]. In line with this hypothesis, a study conducted in Southwest Germany reported the presence of MRSA in nasal swabs taken from calves fed with MRSA-contaminated milk [19]. Moreover, a study involving 20 dairy farms across Germany found a high MRSA-positive test rate in nasal swabs from milk-fed calves (22.7%) compared to nasal swabs sampled from pre-fresh heifers (8.9%) and postweaning calves (9.1%) [20]. The majority of the farms involved in the latter study claimed that they were feeding waste milk to calves [20]. A recent in-depth study used whole-genome sequencing to evaluate the intra-farm transmission of MRSA strains from six German dairy farms [21]. Strikingly, on all six dairy farms closely related MRSA strains could be detected in quarter milk samples and nasal cavities of calves, and in the case of one farm, it was also found on suckers of automatic calf feeders, linking the transmission of MRSA to the feeding process [21]. Thus, MRSA is likely introduced into the calf population by feeding contaminated milk, potentially creating a MRSA reservoir in young calves on dairy farms.

In the first weeks of life, calves are immunologically naive which makes them susceptible to various diseases [22]. To reduce the risk of infection, it is recommended to heat the milk prior to feeding in order to eliminate bacterial pathogens in the milk [22,23]. However, little is known about the heat resistance of MRSA in raw milk and colostrum. The aim of our study was to evaluate whether heat treatment of colostrum and raw milk prior to feeding it to calves will eradicate MRSA in these matrices and to define an optimal temperature–time combination to inactivate MRSA in milk.

## 2. Materials and Methods

### 2.1. Selection of Strains and Preparation of the Inoculum

The MRSA strains (*n* = 17) selected for the thermal resistance experiments originated from bovine mammary quarters, bulk tank milk and nasal swabs (calves and one heifer) and were isolated from 17 dairy farms across eight German federal states [7]. All MRSA strains harbored the *mecA* gene and were resistant against up to seven antimicrobial substances with an overall resistance to β-lactam antibiotics [7]. In addition, the 17 MRSA strains showed large allelic differences, suggesting independent development [7]. The MRSA strains were stored as frozen stocks at −80 °C in bacto peptone broth (Thermo Fisher Scientific, Waltham, MA, USA) containing 43% glycerol. Prior to the thermal challenge, all strains were plated onto Columbia agar with 7% sheep blood (Oxoid GmbH, Wesel, Germany) and incubated at 37 °C for 24 ± 2 h. Liquid cultures were set up in 5 mL brain-heart-infusion (BHI) (Merck, Darmstadt, Germany) and incubated at 37 °C for 24 ± 2 h.

### 2.2. Estimation of the Detection Limit of the Experimental Set Up

To estimate the detection limit of the experimental set up, 10 mL of raw milk was inoculated with the MRSA-1 strain (corresponds to AA1 in [7]) to a level of 100 cfu mL^−1^ or 10 cfu mL^−1^. The inoculated raw milk was immediately added to 90 mL Mueller Hinton broth (Thermo Fisher Scientific Oxoid Ltd., Basingstoke, UK) containing 6% NaCl (Merck, Darmstadt, Germany) and incubated at 37 °C for 24 ± 2 h. Culture aliquots (50 µL) were then plated onto mannitol salt agar (MSA) (Thermo Fisher Scientific Oxoid Ltd., Basingstoke, UK) containing 4 mg L^−1^ cefoxitin (Sigma-Aldrich, Burlington, MA, USA) and incubated at 37 °C for 24 ± 2 h. Presumptive *S. aureus* were identified using MALDI-TOF mass spectrometry (Bruker, Bremen, Germany) as described previously [24]. The experiment was performed twice.

### 2.3. Thermal Challenge

Raw milk was obtained from lactating cows at the German Federal Institute for Risk Assessment, stored cool and used within 10 h. Colostrum of the first milking after calving was collected from an individual cow on a commercial dairy farm in the region. It was stored cool and used within 24 h. The 17 MRSA strains selected for the thermal inactivation experiments were tested individually in raw milk or colostrum. Previously, we had investigated the burden of MRSA in BTM sampled from German dairy farms with a history of MRSA detection and found concentrations of up to 10^3^ cfu mL^−1^ MRSA (unpublished data). Indeed, a few studies in the literature have found 10^5^ cfu mL^−1^ or even higher *S. aureus* concentrations in raw milk drawn from individual animals and BTM [25,26]. For our heat experiment, we therefore added overnight cultures of MRSA to 10 mL of colostrum or raw milk to reach a level of 10^3^ cfu mL^−1^ or 10^6^ cfu mL^−1^. A negative control containing 10 mL of uninoculated colostrum or raw milk was included in the experiment. The inoculated colostrum/raw milk was subsequently immersed in a waterbath and heat treated. A test tube with 10 mL of colostrum/raw milk was used to monitor the temperature, and the incubation time of the samples was measured once the colostrum/raw milk had reached the desired temperature. At defined time intervals, samples were removed from the waterbath and the heat-treated samples were transferred to a flask containing 90 mL Mueller Hinton broth (Thermo Fisher Scientific Oxoid Ltd., Basingstoke, UK) containing 6% NaCl (Merck, Darmstadt, Germany). The raw milk/colostrum control was directly added to the Mueller Hinton broth without prior heating. The flasks were incubated at 37 °C for 24 ± 2 h to allow bacteria to recover from the heat shock. Culture aliquots (50 µL) were then spread onto MSA (Thermo Fisher Scientific Oxoid Ltd., Basingstoke, UK) containing 4 mg L^−1^ cefoxitin (Sigma-Aldrich, Burlington, MA, USA) and incubated at 37 °C for 24 ± 2 h. *S. aureus* were identified via mass spectrometry using a MALDI-TOF Biotyper^®^ system as described above.

### 2.4. Quantification of Thermal Inactivation

In order to quantify the number of MRSA surviving the heat shock, raw milk was inoculated with strain MRSA-1 or MRSA-2 (corresponds to AA1 and BA1 in [7]) to a level of 10^6^ cfu mL^−1^ and heat treated at 60 °C in a waterbath for 10 min. A negative control with 10 mL of uninoculated raw milk was included in the experiment. After the thermal incubation, samples were removed from the waterbath and 100 µL aliquots of the heat-treated milk were spread onto MSA (Thermo Fisher Scientific Oxoid Ltd., Basingstoke, UK) containing 4 mg L^−1^ cefoxitin (Sigma-Aldrich, Burlington, MA, USA). In order to examine the re-growth of MRSA surviving the heat shock, the milk was stored at room temperature (24 °C ± 1 °C) after the heat treatment. Aliquots were removed and diluted in decadal steps in peptone water (Merck, Darmstadt, Germany) after 6 h, 9 h, 24 h and 48 h post treatment. Briefly, 100 µL of the undiluted and the diluted sample (10^−1^ to 10^−5^ dilutions) were then spread onto MSA (Thermo Fisher Scientific Oxoid Ltd., Basingstoke, UK) supplemented with 4 mg L^−1^ cefoxitin (Sigma-Aldrich, Burlington, MA, USA). All plates were incubated at 37 °C for 24 ± 2 h, and presumtive MRSA colonies were counted on plates with no more than 200 total colonies to determine the number of MRSA in the heat-treated samples. The experiment was performed at least three times, and the means with the standard deviations calculated from the biological replicates were plotted.

### 2.5. Statistical Analysis

Statistical analysis was performed to assess the effect of heat treatment on MRSA reduction in raw milk and the growth kinetics of MRSA surviving a heat treatment in raw milk, respectively. To this end, the logarithmic cfu MRSA mL^−1^ values from the independent experiments were entered into Microsoft Excel 2021 (Microsoft Corporation, Redmond, WA, USA) and a paired *t*-test with a one-tailed distribution was applied. Significant statistical differences were considered at *p* < 0.05.

## 3. Results

### 3.1. Heat Inactivation of Low MRSA Levels in Raw Milk and Colostrum

According to the results of the preliminary tests carried out, the estimated detection limit of our thermal challenge experiment is ≤ 10 cfu MRSA mL^−1^. The method is therefore highly sensitive and suitable to detect very low levels of MRSA heat survivors. The vast majority of the MRSA strains (*n* = 16) did not survive 20 min in raw milk heated at 60 °C, and none of the MRSA strains could be detected after an incubation period of 30 min in raw milk heated at 60 °C (Table 1). In addition to raw milk, heat resistance of twelve selected MRSA strains was tested in colostrum. Of the twelve selected MRSA strains, only five survived a heat treatment for 10 min at 60 °C in colostrum, but none of the MRSA strains tested could be detected at later time points during the heat inactivation experiment (Table 1). Thus, the heat resistance of MRSA in the matrices colostrum and raw milk seems to be similar. No MRSA could be recovered from the negative controls, indicating that the raw milk/colostrum was not contaminated with MRSA prior to incoculation.

### 3.2. Heat Inactivation of High MRSA Levels in Raw Milk

A heat treatment at 60 °C for up to 30 min did not completely inactivate high levels of MRSA in raw milk (Table 2). Moreover, MRSA could be detected after an incubation period of 30 min in raw milk heated at 65 °C (Table 2). Therefore, the temperature was further increased to 70 °C. Most of the MRSA strains (n = 12) survived 10 min in raw milk heated at 70 °C, and three MRSA strains were also recovered after 20 min of incubation at 70 °C (Table 2). A heat treatment for at least 30 min at 70 °C was required to completely inactivate all MRSA in highly inoculated raw milk (Table 2).

### 3.3. Quantifcation of MRSA Heat Inactivation in Highly Inoculated Raw Milk

Our analysis of two selected MRSA strains revealed that already a short heat treatment for 2 min at 60 °C reduces the number of viable MRSA cells in raw milk by 3–4 logarithmic levels (Figure 1a). The differences in MRSA counts before and after the 2 min heat treatment were statistically significant (MRSA-1: *p* = 0.0002 and MRSA-2: *p* = 0.0004). However, when the inoculated raw milk was stored at room temperature after the heat shock, re-growth of both MRSA strains was detected 24 h post heat treatment (Figure 1b). Statistical analysis revealed that the increase in MRSA 24 h post heat treatment was significant (MRSA-1: *p* = 0.0004 and MRSA-2: *p* = 0.0004). In contrast, the MRSA counts in the raw milk 9 h post heat treatment were not significantly different from the MRSA counts after the 10 min heat shock at 60 °C (Figure 1b). 

## 4. Discussion

The feeding of waste milk on dairy farms provides a cost-effective alternative to milk replacer. However, waste milk may be contaminated with antimicrobial residues and pathogenic or other bacteria, originating either from the mammary gland or from the milking environment. Milk is highly nutritious and provides an ideal environment for bacterial growth [27]. As a consequence, waste milk may cause disease in calves due to the intake of pathogenic bacteria [28]. Moreover, antimicrobial residues may support the selection of resistant bacteria [18]. In addition, calves that become colonized or infected via exposure to contaminated waste milk may contribute to the spread of antibiotic resistance genes within the herd. In order to break the cycle of transmission of these bacteria and generate a safe product to feed calves, control measures such as heating the raw milk prior to feeding are recommended on dairy farms [23]. However, the temperature and exposure time needed for inactivation differs among bacteria and may also effect the quality of the milk which hampers clear guidelines [29]. In addition, heating milk above 60 °C may destroy immunoglobulins in the milk that are vital for the calf’s development [30]. Thus, temperatures above 60 °C are not desirable for milk destined for feeding. Newborn calves are fed with colostrum shortly after birth. Colostrum is the first milk produced by mammals after delivery and is compositionally distinct from raw milk. Colostrum has a higher protein and fat content, but significantly less lactose compared to mature milk [31]. The unique composition of colostrum is important for the development of the calf, in particular the passive transfer of immunoglobulins from colostrum is needed to protect the calf against infections [32]. McMartin S. et al. (2008) showed that heating colostrum at 63 °C resulted in a substanial decrease in immunoglubulin G levels [33]. In contrast, no differences in immunoglubulin G levels between pre-heat-treated and post-heat-treated colostrum samples were observed at 60 °C [33]. Moreover, heat treatment of colostrum in a commercial batch pasteurizer at 60 °C for 60 min preserved the immunoglubulin G fraction, although this may depend on the quality of the colostrum, as a reduction was observed, particularly in samples with high initial immunoglubulin G concentrations [34].

Little is known about the heat resistance of MRSA in colostrum or raw milk since most studies in the literature were performed with methicillin-sensitive *S. aureus* (MSSA) strains in divergent matrices. Interestingly, depending on the strain and the matrix used in the experiment, substantial differences in *S. aureus* heat sensitivity have been described. For example, McKay et al. (2008) evaluated the decimal reduction time (D-value) at 56 °C of nine *mecA*-negative *S. aureus* strains in ultra-heat-treated whole milk and reported values between 10.8 and 20.1 min [35]. An earlier study investigated the heat sensitivity of two toxin-producing MSSA strains in skim milk and reported significantly lower D_56_ values of 9.2 and 9.8 min, respectively [36]. Kennedy et al. (2005) examined the thermal resistance of a five *S. aureus* strains containing cocktail in sterile tryptic soy broth and described D-values at 60 °C of 4.8 to 6.5 min, independent of whether the cells were pre-chilled before the heat treatment or not [37]. In comparison, our experiments revealed a more heat-sensitive phenotype of MRSA, in which a heat treatment for 4–6 min at 60 °C led to a 4–5 logarithmic reduction, followed by a constant count of heat-resistant MRSA until the end of the heat treatment (10 min). Remarkably higher D_60_-values were reported by Amado et al. (2014) when studying the effect of cattle feed structure in the thermal inactivation of *S. aureus* [38]. According to the latter study, 66.3 min, 35.7 min and 26.9 min were required to achieve one log reduction in *S. aureus* at 55 °C, 57.5 °C and 60 °C, respectively. Interestingly, the authors also observed a tailing effect when analyzing *S. aureus* heat inactivation curves, suggesting the presence of a highly heat-resistant subpopulation [38]. Furthermore, a study conducted by Li et al. (2005) evaluated the thermal resistance of *S. aureus* in high solids liquid egg mixes and reported D_64_ values of 1.3 to 1.8 min [39]. At least three studies in the literature described an unusual high heat-resistant phenotype of *S. aureus*. Montanari et al. (2015) studied the heat inactivation of staphylococci including *S. aureus* in BHI and found that all strains tested survived a heat treatment for 20 min at 80 °C. Moreover, the authors showed that during the first 2–4 min at 80 °C, the number of staphylococci dropped six log units, but afterwards remained constant until the end of the heat treatment [40]. A second study conducted by Yehia et al. (2019) reported an enterotoxin-producing *S. aureus* surviving a heat treatment for 2 min at 90 °C in BHI [41]. In a recent work by Yehia et al. (2020), the authors isolated 10 MRSA strains from camel milk and demonstrated that all strains were highly heat resistant with 3/10 strains even surviving 90 s at 90 °C in BHI [42]. In accordance with these high temperature treatment studies, in our study too 12/17 MRSA strains survived a heat treatment in raw milk for 10 min at 70 °C, providing high levels of MRSA (10^6^ cfu mL^−1^), and 3/17 MRSA strains were even recovered after heating the raw milk for 20 min at 70 °C.

Our study provides thermal inactivation data for MRSA in the matrices colostrum and raw milk. Our data show that the termperature–time requirements to fully inactivate MRSA in raw milk depends on the initial contamination level. The results suggest that a holding time of 30 min in raw milk heated at 60 °C is sufficient to fully eradicate low levels of MRSA (10^3^ cfu mL^−1^) in colostrum and raw milk. Despite the compositional differences between colostrum and raw milk, we did not detect significant differences in the heat sensitivity of MRSA in these two matrices in our study. However, highly MRSA-contaminated raw milk (10^6^ cfu mL^−1^) requires temperatures of at least 70 °C to completely inactivate MRSA. Using a quantitative approach, we demonstrate that even though heating raw milk at 60 °C may not eliminate all viable MRSA in the raw milk, it reduces the number by several log levels and therefore may lower the risk for MRSA transmission. Overall, our data provide relevant information on the heat resistance of MRSA in colostrum and raw milk that may help to reduce the spread of MRSA on dairy farms.

## 5. Conclusions

The tested MRSA strains slightly differed in their heat resistance. Nevertheless, we conclude that heat treatment of colostrum and raw milk at 60 °C allows for a substantial reduction in MRSA in the colostrum and raw milk (MRSA-1: *p* = 0.0002 and MRSA-2: *p* = 0.0004). However, depending on the MRSA concentration, it may not be sufficient to eliminate all viable MRSA. Heat-treated milk should therefore be fed to the calves as soon as possible to avoid re-growth of viable cells. Unfortunately, it is not clear whether the observed reduction in MRSA is sufficient to preclude infection or colonization of calves. This would require further study and feeding experiments.

## Figures and Tables

**Figure 1 animals-13-03549-f001:**
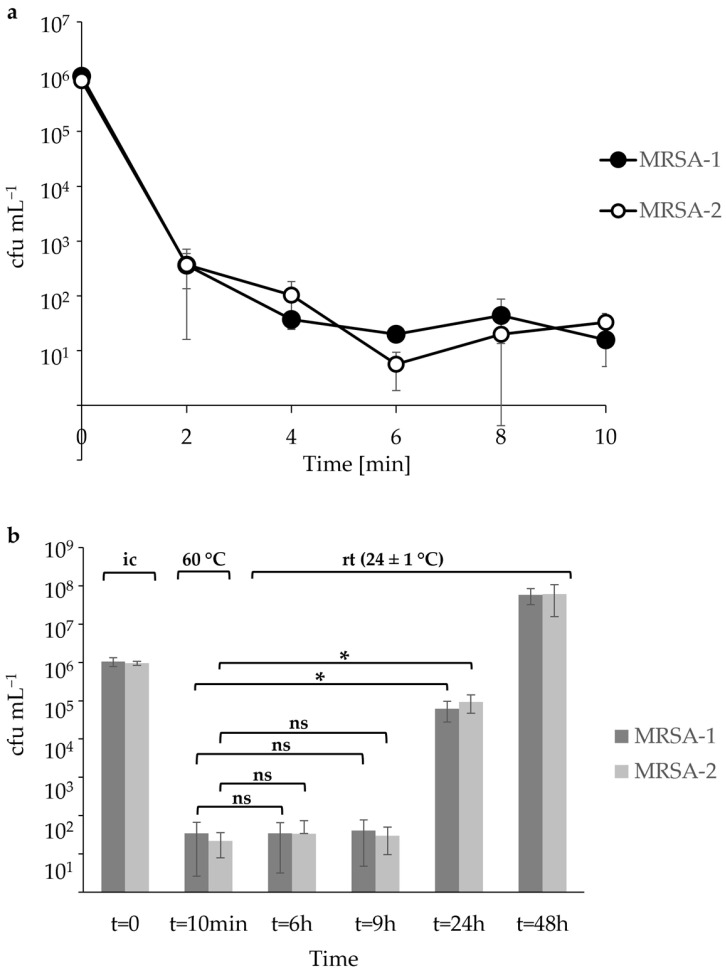
(**a**) Quantification of thermal inactivation of two Methicillin-resistant *Staphylococcus aureus* (MRSA) strains in raw milk at 60 °C. Raw milk was inoculated with MRSA to a level of 10^6^ cfu mL^−1^ and heat treated at 60 °C. At defined time points, the heat-treated raw milk was sampled and the cfu mL^−1^ were determined by plating on selective agar plates as described in Material and Methods. (**b**) Growth of two MRSA strains in raw milk following a heat shock for 10 min at 60 °C. Raw milk was inoculated as described in (**a**), heat treated for 10 min at 60 °C and then stored at room temperature (rt) for 48 h. Immediately after and at defined time points post heat treatment, aliquots were sampled and the cfu mL^−1^ were determined by plating on selective agar plates. For statistical analysis, a paired Student’s *t*-test was performed, and statistically significant differences with *p*-values below 0.05 are indicated with an asterisk (*). (ns) = not significant, (ic) = inoculum.

**Table 1 animals-13-03549-t001:** Thermal resistance of low Methicillin-resistant *Staphylococcus aureus* (MRSA) levels (10^3^ cfu mL^−1^) in colostrum and raw milk heated at 60 °C. Colostrum or raw milk was inoculated with MRSA to a level of 10^3^ cfu mL^−1^ and heat treated at 60 °C in a waterbath. At defined time intervals, the heat-treated colostrum or raw milk was sampled and the recovery of MRSA was qualitatively investigated via selective enrichment broths as described in Material and Methods. *S. aureus* colonies were confirmed using a MALDI-TOF Biotyper System.

60 °C Raw Milk				60 °C Colostrum		
Time (min)	10	20	30	10	20	30
x/y *	9/17	1/17	0/17	5/12	0/12	0/12

* (x/y) = number of *S. aureus* strains detected/number of *S. aureus* strains tested.

**Table 2 animals-13-03549-t002:** Thermal resistance of high Methicillin-resistant *Staphylococcus aureus* (MRSA) levels (10^6^ cfu mL^−1^) in raw milk heated at 60 °C, 65 °C or 70 °C. Raw milk was inoculated with MRSA to a level of 10^6^ cfu mL^−1^ and heat treated at 60 °C, 65 °C or 70 °C in a waterbath. At defined time intervals, the heat-treated milk was sampled and the recovery of MRSA was investigated qualitatively via selective enrichment broths as described in Material and Methods. *S. aureus* colonies were confirmed using a MALDI-TOF Biotyper System.

	60 °C Raw Milk	65 °C Raw Milk	70 °C Raw Milk		
Time (min)	30	30	10	20	30
x/y *	2/2	2/2	12/17	3/17	0/17

* (x/y) = number of *S. aureus* strains detected/number of *S. aureus* strains tested.

## Data Availability

The data presented in this study are available in this manuscript or on request from the corresponding author.

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
