# Peer review of "Heat Inactivation of Methicillin-Resistant Staphylococcus aureus Strains from German Dairy farms in Colostrum and Raw Milk"

_animals, 2023, doi:10.3390/ani13223549_

Round 1
Reviewer 1 Report
Comments and Suggestions for Authors
Dear editor and authors,
thank you for the opportunity to review interesting manuscript: Heat inactivation of methicillin-resistant Staphylococcus aureus strains from German dairy farms in colostrum and raw milk. The manuscript has potential for application and meets the requirements for publication in the Animals. The authors have made several mistakes, which do not diminish its quality. I have the following comments on some parts:
The title of the article is suitable.
Abstract
The abstract reflects the aim of the study with the achieved results.
Lines 25 – 26: Heating colostrum and raw milk above 60 °C may destroy 25 immunoglobulins which are crucial for the calf’s health. It is a speculative statement and is a pity that the authors did not include in the study the thermal degradation of immunoglobulins during heating of MRSA.
Introduction
Lines 40 – 42; 45 – 47 and 52 -55: The citations 12, 32 and 33 are old. After the sentences, I recommend adding more recent citations: https://doi.org/10.3390/ani12040470
https://potravinarstvo.com/journal1/index.php/potravinarstvo/article/view/905
Materials and methods
Line 82: Specify, how many MRSA strains were selected. MRSA must be confirmed first phenotypically, the genotypically. I hope authors tested by PCR only strains that showed phenotypic resistance. If not, then in the write up, it should clearly state that they detected phenotypic resistance only.
Results
Is missing statistical expression with commentary in figure 1.
Conclusion
The conclusion is clear and reflects the most important findings.
Comments on the Quality of English LanguageMinor editing of English language required.
Reviewer 2 Report
Comments and Suggestions for Authors
Global comment
In this paper, the authors used 17 MRSA strains previously identified and obtained from dairy farms. These MRSA with different concentrations were used to inoculate or contaminate colostrum and raw milk that was submitted for heat treatment [60 °C or above]. The presence of these MRSA strains was tested 10min, 20, and 30 min after heat treatment, as well as the bacterial growth of milk stored at room temperature.
The pasteurization of colostrum is an increasing treatment used in farms to eliminate pathogenic agents and antibiotic-resistant bacteria, as well as mitigate the transmission of these for young animals and humans. However, data about specific antimicrobial resistance is limited. These results are interesting for the scientific community and address a global concern.
In general, the manuscript is well-written with scientific quality. This study allows us to study measures to mitigate antibiotic-resistant Staphylococcus aureus through the route of administration of colostrum for animals. The methodological approach and results are accurate but it can be improved, therefore I suggest some revisions.
Comments
Materials and Methods
Line 87: Please, confirm the concentration of glycerol.
Line 94: Why did the authors choose a level of 100 cfu mL-1 or 10 cfu mL-1 to contaminate the raw milk?
The raw milk and colostrum used in this manuscript, were tested first for the presence of MRSA? What is the origin of colostrum and raw milk? How were these collected and how much time were collected before?
The 17 MRSA strains inoculated in colostrum or raw milk were tested individually? Please clarify.
Line 124. Confusing. In the thermal inactivation were 17 MRSA isolates, all phenotypic and genotypic different, or were 17 MRSA only corresponding to the phenotypic and genotypic of AA1 and BA1?
Results
Line 145-149. Revised the initial part of the sentence. It should not start with numbers as ‘16/17’. Please, check the entire manuscript.
Line 162. And colostrum results with high MRSA levels, was it tested?
Comments on the Quality of English LanguageMinor editing of English language required.
Reviewer 3 Report
Comments and Suggestions for Authors
Dear author,
The topic you have addressed is quite relevant at the moment, knowing the issues are faced in cattle farms with youth management.
I am convinced your manuscript will be published soon, however, there are still a few revisions to address.
Abstract: please rewrite lines 28-29, one may think there are different types of milk to be fed to valves... 1st, the 60deg treated then an other type, heat treated.
Introduction: try to shorten it a bit to become more concise. Also, throughout all the document, you must number the reference list in the order of appearance, not using the alphabetical order. So, you have to reposition the citations in the references list and to give them the appropriate number in the text. See the instruction for authors, here: Animals | Instructions for Authors (mdpi.com)
2. Materials and Methods
Please add a 2.5 section to describe the statistical analysis you have performed on experimental data, stating the applied method, number of repetitions/readings in the database, the software etc.
I suppose you have run at least one simple Student test or a Fisher to compare between the methods of thermal treatment and control. Therefore, please add these data and also in the results/discussion sections put the corresponding p values. In the conclusions section, you state about a substantial reduction... This statement means you have run analysis of variance or other test to compare the significance level of the differences between means.
Please put these results or run the comparisons on the data base and input the outcomes in the paper.
Also, please revise your English all throughout the paper.
Good luck!
Comments on the Quality of English Language
English could be improved. It is recommended to be proofread by a native speaker or by a scholar with better fluency in English.
Round 2
Reviewer 3 Report
Comments and Suggestions for Authors
Dear authors,
Thank you for your efforts in improving the quality and concision of your paper.
Best regards!